# Ordering of Rods near Surfaces: Concentration Effects

**Dora Izzo** 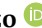

Instituto de Física, Universidade Federal do Rio de Janeiro, C.P. 68528, Rio de Janeiro 21941-972, Brazil; izzo@if.ufrj.br

**Abstract:** We study the orientation of rods in the neighborhood of a surface. A semi-infinite region in two different situations is considered: (i) the rods are located close to a flat wall and (ii) the rods occupy the space that surrounds a sphere. In a recent paper we investigated a similar problem: the interior of a sphere, with a fixed concentration of rods. Here, we allow for varying concentration, the rods are driven from a reservoir to the neighborhood of the surface by means of a tunable chemical potential. In the planar case, the particle dimensions are irrelevant. In the curved case, we consider cylinders with dimensions comparable to the radius of curvature of the sphere; as they come close to the surface, they have to accommodate to fill the available space, leading to a rich orientational profile. These systems are studied by a mapping onto a three-state Potts model with annealed disorder on a semi-infinite lattice; two order parameters describe the system: the occupancy and the orientation. The Hamiltonian is solved using a mean-field approach producing recurrence relations that are iterated numerically and we obtain various interesting results: the system undergoes a first order transition just as in the bulk case; the profiles do not have a smooth decay but may present a step and we search for the factors that determine their shape. The prediction of such steps may be relevant in the field of self-assembly of colloids and nanotechnology.

**Keywords:** rods; curved surface; Potts

## 1. Introduction

In the past fifty years, a great deal of attention has been devoted to understanding how liquid crystals are ordered in small cells. Applications in technology have pushed forward the interest in such systems: optical devices with specific properties depend on the anchoring conditions of the mesogens to the surfaces [1]; coating spherical colloidal particles with thin layers of liquid crystals in specific ways provides flexibility for tuning directional interactions between colloids [2]. In that scenario, particles are of the order of a few nanometeres, much smaller than their confining volumes, with sizes in the micrometer range: a description in terms of continuum theories [3] is completely satisfactory.

More recently, larger particles, commonly named macroscopic liquid crystals, have also captured attention. Examples of such structures are liquid crystals formed by viruses [4,5], filamentous biopolymers [6] and carbon nanotubes [7,8], with dimensions of the order of hundreds of nanometers. Not only is confinement a relevant issue, but also self-assembly of colloids mediated by large particles is a new problem to be investigated. From a theoretical point of view, continuum theories break down and one looks to understand how these large particles accommodate finding their optimal conformation when mutual packing and alignment to boundaries compete at the same length scale. Some theoretical descriptions have addressed such problems using Monte Carlo simulations and molecular dynamics [9–12].

Our system consists of a colloidal liquid crystal represented by a set of rod-like particles. They are driven from a reservoir by means of a chemical potential and we are primarily interested in describing

how these rods self-organize around a curved surface. The concentration is not fixed and vacancies come into play, leading to *annealed* disorder. Excluded volume interactions are considered and assist organization because the particle dimensions are sizeable with respect to the surface curvature radius, typically on the order of 1/10 (as in the case of filamentous bacteriophage fd-viruses in microchambers); Figure 1 illustrates this situation.

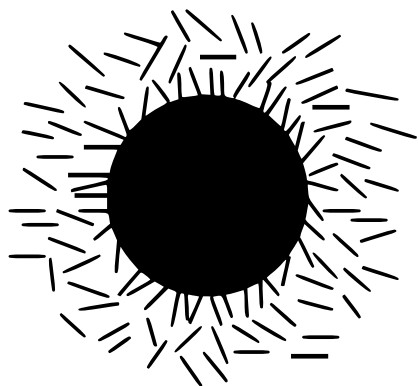

**Figure 1.** Representation of the distribution of rods around a sphere; here, we show a cross section of the three-dimensional system. For large particle concentrations, the rods are forced to align around the sphere; this alignment decreases towards the bulk.

For a given value of a chemical potential (large enough to guarantee minimal occupancy just next to the sphere), the orientation profile can be understood as follows: (i) just outside the sphere there is enough room to accommodate a given number of particles, anchored homeotropically; (ii) beyond this region, in the upper rows, more room is available, which amounts to a decrease in the excluded volume interactions and therefore a reduction in ordering; (iii) for rows further away from the sphere, the "squeezing" effect vanishes and an isotropic configuration sets in.

We model this system in terms of a lattice surrounding a sphere. Particles (rods) occupy the lattice sites, but vacancies may occur. We follow Oliveira and Figueiredo Neto [13] and associate the orientation of each rod to a microscopic state of the three-state Potts model and solve it on a lattice. Nevertheless, the lattice description of the curved three-dimensional system is unfeasible (discussed in the next session) and we are led to solve the two-dimensional version of the problem.

The system described above is the main subject of this paper. Nevertheless, we will also address a related problem: the effect of *quenched* disorder. In this case, it is more realistic to consider the system studied in a previous work [14], in which large rods are located inside a sphere and all lattice sites are occupied. We predict the behavior of this system in terms of a qualitative discussion.

This paper is organized as follows: in Section 2, we define the two-dimensional version of the problem and describe the mean-field approach for the three-state Potts model with annealed disorder; in Section 3, the results are presented, first for a simpler problem, the planar surface (PS) case, and then for the curved surface (CS) case; these results are discussed in Section 4, where we comment on the effect of introducing quenched disorder in a similar system; conclusions are left for Section 5.

## 2. The Problem

### 2.1. The Two-Dimensional Version

We model the colloidal liquid crystal as a system of rods in a lattice. A single rod is located on a lattice site but not all sites must be occupied. Because it is not possible to embed a curved object into a cubic lattice, we reduced our problem to two-dimensions redefining the lattice structure and the rods' geometry as follows.

The three-dimensional version of the planar surface case (PS) consists of a cubic lattice bounded by a flat surface. The analogous two-dimensional problem is just a square lattice bounded by a line.

In order to study the two-dimensional analogue of the curved surface case (CS), the sphere is replaced by a disk. We choose a generic cross section of the three-dimensional space that surrounds a sphere with a planar lattice network that represents the symmetry of the problem as illustrated in Figure 2.

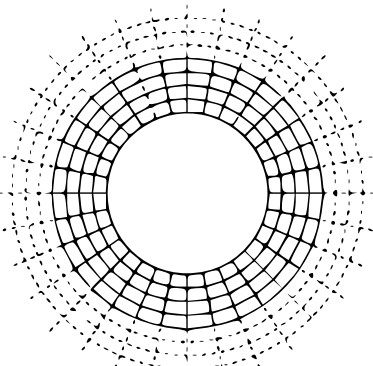

**Figure 2.** The network of lattice sites around a disk.

Accordingly, the rods must be replaced by rectangles with linear dimensions $b$ and $L$ ($b < L$). Every geometric center is fixed on a lattice site and rotates on the lattice plane. Because only excluded volume interactions are considered, the alignment of a given rod may be constrained only by its nearest neighbors' orientation. Next, we discuss the geometric constraints and show that, in the CS case, the geometry is responsible for an extra channel of disorder towards the bulk.

Consider a planar lattice such as the one in Figure 2; let $R$ be the radius of the disk and $N$ the number of lattice sites by row. Accommodating homeotropic anchored rods around the circle requires: $R > N\rho(0)b$. Towards the bulk, the increase in distance between nearest neighbor rods amounts to a decrease in excluded volume interactions (mutual alignment) and rotation is gradually allowed; at $\bar{r} = N\rho(\bar{r})L$, excluded volume interactions finally die out, which characterizes the "bulk". Therefore, the region of interest is $R < r < \bar{r}$.

### 2.2. The Three-State Potts Model with Annealed Disorder

We consider a plane (either PS or CS) and use the mapping proposed by Oliveira and Figueiredo Neto [13]. Because we aim to describe occupancy and ordering, we need two order parameters.

The concentration profile is determined by a set of order parameters $\{\rho_\ell\}$ associated with the fraction of occupied sites in the system. $\rho_\ell$ is an average over a microscopic parameter $\eta_\ell$ on the $\ell$th row: $\rho_\ell \equiv \langle \eta_\ell \rangle_0$.

The directional order in each row is described by another set of order parameters defined as follows. The orientation of each rod refers to the surface director; in the PS case, the surface is a straight line; in the CS case, it is a circle. We use the three-state Potts model to describe the system: one of the three states corresponds to alignment perpendicular to the surface (homeotropic), whereas the other two are degenerate and refer to alignment in the other orthogonal directions parallel to the surface (planar). The state of ordering is thus described by the set $\{q_\ell\}$, obtained by an average over the microscopic parameters $\eta_\ell$ and $s_\ell$ on the $\ell$th row: $q_\ell = \langle \eta_\ell.s_\ell \rangle_0$, where the variable $s_\ell$ describes the microscopic directional state on row $\ell$. In fact, $q_l$ specifies the corresponding averaged moment of quadrupole with respect to a homeotropic alignment in the rod frame of reference $\Lambda_\ell$ [15]:

$$\Lambda_\ell = \begin{bmatrix} -q_\ell/2 & 0 & 0 \\ 0 & -q_\ell/2 & 0 \\ 0 & 0 & q_\ell \end{bmatrix}. \tag{1}$$

In a mean-field description of the annealed three-state Potts model, the expression for the energy per column of a given plane, either in the PS or in the CS cases, is given by [16]:

$$\mathsf{F} = -\mu \sum_{\ell=0}^{\infty} \rho_l - \sum_{\ell=0}^{\infty} \epsilon_\ell q_\ell q_{\ell+1} - \sum_{\ell=0}^{\infty} \epsilon_\ell (q_\ell)^2 + \frac{k_B T}{3} \sum_{\ell=0}^{\infty} \left[ \ln(\rho_\ell + 2q_\ell) + 2\ln(\rho_\ell - q_\ell) - 3\ln 3(1 - \rho_\ell) \right] . \quad (2)$$

The index $\ell$ labels the rows: $\ell = 1$ refers to the first row, next to the surface; $\rho_0$ and $q_0$ fix the anchoring conditions as specified below. Each site has two nearest neighbors on the same row and one nearest neighbor on the row just above it. $\mu$ is a common chemical potential that drives particles from the reservoir and $\epsilon_\ell$ is the interaction energy between a rod on row $\ell$ and its nearest neighbors.

Following the general procedure of the mean-field approach, we minimize expression (2) with respect to $\rho_\ell$ and $q_\ell$, which yield the recurrence relations:

$$\begin{cases} -\mu + \frac{k_B T}{3} \left[ \ln(\rho_\ell + 2q_\ell) + 2\ln(\rho_\ell - q_\ell) - 3\ln 3(1 - \rho_\ell) \right] = 0, \\[2mm] -\epsilon_\ell q_{\ell+1} - \epsilon_{\ell-1} q_{\ell-1} - 2\epsilon_\ell q_\ell + \frac{2}{3} k_B T \ln \frac{\rho_\ell + 2q_\ell}{\rho_\ell - q_\ell} = 0. \end{cases} \quad (3)$$

This set is solved numerically under constraints in the bulk and on the surface as follows. In the bulk, $\rho_\infty$ and $q_\infty$ are obtained from Equation (3), by fixing $\rho_\ell = \rho_{\ell+1} = \rho_{\ell-1} = \rho_\infty$ and $q_\ell = q_{\ell+1} = q_{\ell-1} = q_\infty$, leading to:

$$\begin{cases} \rho_\infty = \frac{(1-\rho_\infty)^3 \exp 3\mu/k_B T}{(\rho_\infty - q_\infty)^2} - 2q_\infty, \\[2mm] q_\infty = \frac{k_B T}{6\epsilon_\infty} \ln \frac{\rho_\infty + 2q_\infty}{\rho_\infty - q_\infty}. \end{cases} \quad (4)$$

On the surface, we must specify the anchoring conditions: $\rho_0$ and $q_0$. Equation (3) set the limits for stability:

$$\begin{cases} 0 < \rho_\ell < 1 & \text{and} \\[1mm] -1/2 < q_\ell < \rho_\ell < 1. \end{cases} \quad (5)$$

The first of the above equations is trivial; the second shows that, for a given occupancy, the upper and lower bounds of $q_0$ correspond to homeotropic and planar anchoring, respectively, while $q_0 = 0$ is associated with a disordered state on the surface.

## 3. Numerical Procedure and Results

We iterate the equations presented above; three hundred layers are enough to obtain a full picture of the profile. The region far from the surface, but still under the effect of the chemical potential, shal be termed "bulk"; the particle reservoir, free from the chemical potential, is not shown in the profiles below. In what follows, we use reduced non-dimensional parameters: the temperature $\theta \equiv k_B T / \epsilon$ and the chemical potential $a \equiv \mu / \epsilon$.

Here, similarly to what was reported previously [14], the specific values of the homeotropic anchoring conditions $\rho_0$ and $q_0$ affect only the immediate vicinity of the surfaces (not shown here); the profiles are determined primarily by the bulk configuration. Moreover we notice that, for $\mu = 0$, the high temperature values of $\rho$ and $q$ are 0.75 and 0.0, respectively.

We start by presenting the results for the PS case and then turn to the CS case.

*3.1. The Planar Surface (PS) Case*

The existence of a first order transition is well known in analogous three-dimensional systems. Figure 3 shows the phase diagram describing the orientational ordering transition in the case of non-confinement: regions *I* and *II* correspond to the ordered (nematic) and isotropic phases, respectively.

In the presence of a surface, the first order phase transition persists; we find that ordering transition. for $\mu = 0$, the discontinuous transitions set in for $1.61 < \theta < 1.72$, and occur for $\rho$ and $q$

simultaneously. This is not the case for $\mu \neq 0$, where $\rho$ and $q$ have transitions in different windows of temperature. Figure 4 shows their coexistence profiles for $\mu = 0$ in the beginning ($\theta = 1.62$) and in the end ($\theta = 1.69$) of the first order transition window; they show a persistence step in the disordered solution reminiscent from the homeotropic boundary condition on the surface; the steps are wide for $\theta = 1.62$, which is closer to the ordered end of the transition, while, for $\theta = 1.69$, the steps almost disappear. We emphasize that $\theta$ is associated with the bulk degree of order, which means that the step width is determined by the bulk configuration.

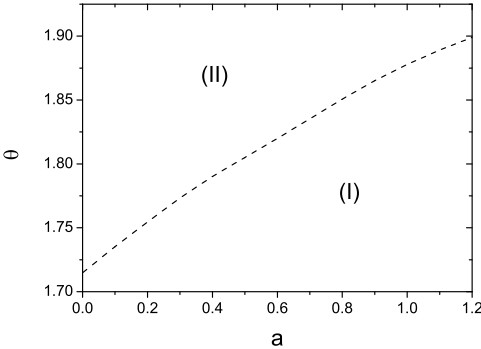

**Figure 3.** Phase diagram for the bulk ordering transition, described by the non-dimensional chemical potential and temperature. The nematic and isotropic phases correspond to regions *I* and *II*, respectively. The dashed line represents the discontinuous phase transition.

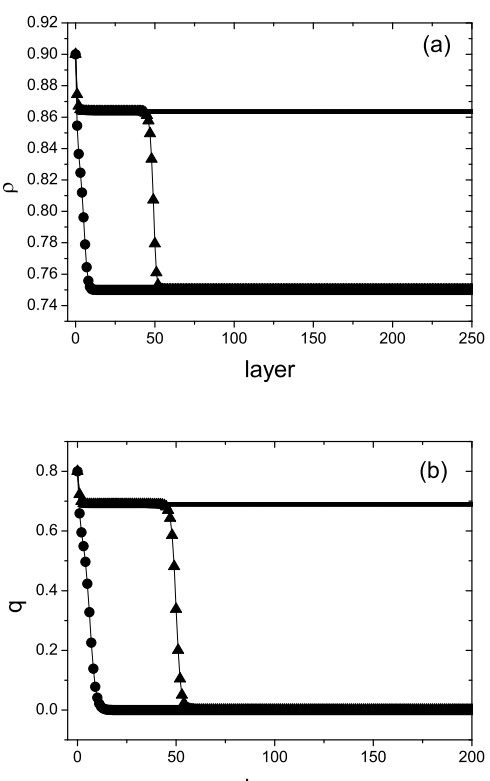

**Figure 4.** PS case profiles for $a = 0$: (**a**) occupancy and (**b**) orientational order parameter. Ordered solutions for $\theta = 1.62$ (continuous lines) and disordered solutions (circles and triangles). Circles: $\theta = 1.69$, triangles: $\theta = 1.62$, corresponding to the high and low temperature ends of the first order window.

Next, we study the phase coexistence for fixed temperature and varying chemical potential. Similarly to what was reported above, the mean-field solution predicts a window of coexistence, rather than a single point. Figure 5 illustrates this aspect of the transitions: we fix $\theta = 1.69$ and vary $a$. In Figure 5a, we notice that $\rho$ changes discontinuously; this jump occurs somewhere between $a \simeq 0.20$ and $a \simeq 0.82$. In Figure 5b, a similar jump is observed in $q$, within exactly the same $a$ range.

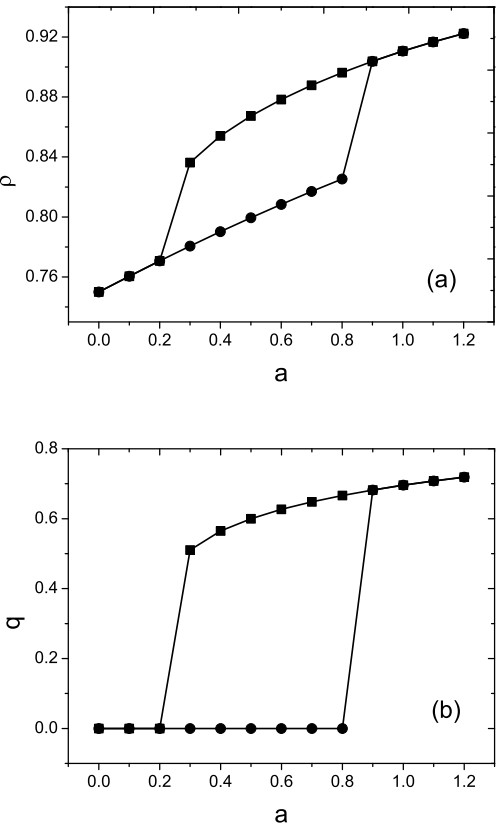

**Figure 5.** PS case. First order transition as a function of $a \equiv \mu/\epsilon$ for fixed $\theta = 1.69$. (**a**) occupancy and (**b**) orientational order parameter. Circles: disordered solution, squares: ordered solution. Lines are just guides to the eye.

An important issue to be investigated is the influence of $\mu$ on the steps shape. Figure 6 illustrates this behavior: the steps go further into the bulk with increasing $\mu$. We also verified that, rather than increasing slowly, at $a = 0.07$, the steps disrupt into full plateaus (both for $\rho$ and $q$), a manifestation of the first order transition. Concerning the steps' height, the $\rho$ profile is more sensitive to the increase in $\mu$ than the $q$ profile.

These results show that the width of the nematic film increases with $a$, lending strong support for complete orientational wetting transition in the limit $a \to a_c$, where $a_c = 0.07$. Therefore, we studied the increase of the adsorption $\Gamma$ for a system of $N$ layers, defined as:

$$\Gamma = \frac{1}{N} \sum_{\ell=1}^{N} [\rho_\ell - \rho_\infty] \tag{6}$$

as a function of $(a - a_c)/a_c$, shown in Figure 7, confirming the complete orientational wetting scenario.

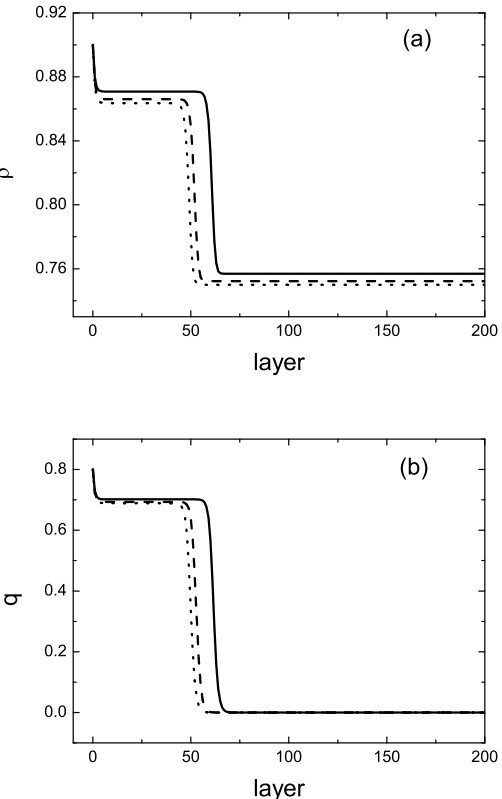

**Figure 6.** PS case. Effect of increasing chemical potential on profiles step for $\theta = 1.62$. (**a**) occupancy and (**b**) orientational order parameter. Dotted line: $a = 0.00$, dashed line: $a = 0.02$ and continuous line $a = 0.06$.

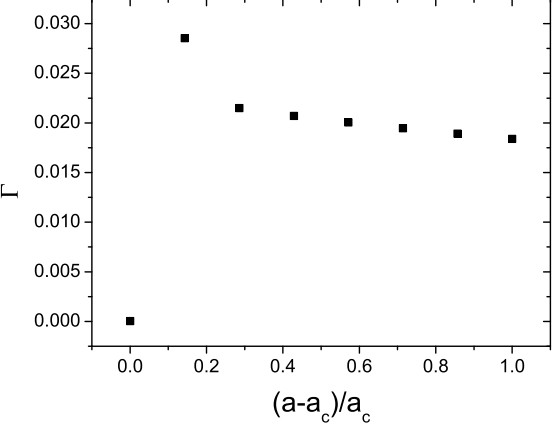

**Figure 7.** PS case. Adsorption $\Gamma$ as a function of the difference between the reduced chemical potentials below the transition and at the transition ($\theta = 1.62$).

We also checked the effect of imposing planar anchoring. Figure 8 shows profiles that illustrate the situation: in this case, the boundary condition on the wall is strongly planar ($q(0) = -0.49$, $\rho(0) = 0.9$), $a = 0$ and we considered two temperatures in the vicinity of the first order transition, $\theta = 1.60$ and $\theta = 1.70$. For the lower temperature (below the transition), the first layers show planar ordering and eventually reach an condition; for the higher temperature (above the transition), the first layers show also planar ordering that decay into an isotropic state towards the bulk. It seems that no long range

planar ordering can be reached throughout the sample: this is because the mean-field approach for Potts models always favours one direction.

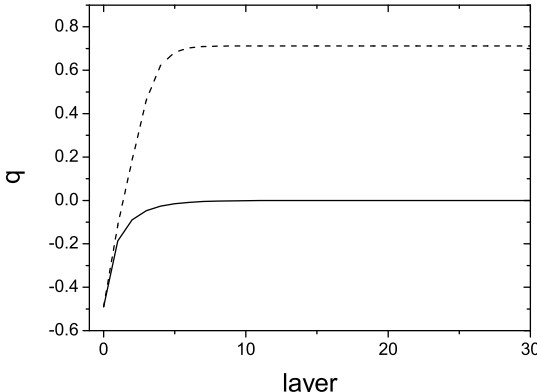

**Figure 8.** PS case. Profile for the orientational order in the case of strongly planar anchoring, $a = 0.00$. Continuous line: $\theta = 1.60$, just below the transition; dashed line: $\theta = 1.60$, just above the transition.

Next, we turn to the more interesting situation: the curved surface case.

### 3.2. The Curved Surface (CS) Case

Here, we show the results for decreasing interactions towards the bulk: we fix the interactions values on the surface ($\theta_s \equiv \epsilon_1 / k_B T$) and in the bulk ($\theta_b \equiv \epsilon_\infty / k_B T$); we may also impose the interaction decrease rate.

Here, similarly to the last case, first order transitions are obtained within a window of parameters: both for the bulk temperature at fixed chemical potential and for the chemical potential at fixed bulk temperature (not shown here).

In the vicinity of the surface, for sufficiently strong interparticle interactions (low $\theta_s$), an ordered configuration sets in. In order to describe how the system evolves to a less ordered, or even disordered state in the bulk, we vary the interaction decrease rate ($\theta$ increase rate). Figure 9 shows profiles for $\epsilon_\ell \sim, \ell^{-1}$ and $\epsilon_\ell \sim \ell^{-0.7}$, for $\theta_s = 1.69$, $\theta_b = 1.80$ and $a = 1.0$. The occupancy (Figure 9a) decreases to a finite value in the bulk, whereas the orientational order parameter (Figure 9b) decreases to zero (isotropic state). Both order parameter profiles reach the corresponding bulk values presenting a step associated with an ordered region close to the surface; the width of the steps increases as the interaction decay rate becomes smoother.

The dependence of $\epsilon_\ell$ with $\ell$ describes the alignment strength between a rod in row $\ell$ and its neighbors. Let $R$ be the circle (surface) radius, $a$ the distance between adjacent rows, $d_\ell$ and $d_0$ the distance between neighboring sites on row $\ell$ and on the surface, respectively. Then, $(R + \ell.a)/N \simeq d_\ell$. If $\epsilon \sim \ell^\gamma$, where $\gamma$ is a generic exponent, then $\epsilon(d_\ell) \sim (d_\ell/d_0 - 1)^{-\gamma}$. This means that the alignment strength $\epsilon$ depends on a non-dimensional distance between rods $\bar{d}$ like $(\bar{d} - 1)^{-\gamma}$.

Another aspect regarding steps is that they occur just in situations where the bulk degree of disorder is not too high. As the bulk temperature increases, the step width decreases, up to a situation where the bulk disorder washes out the step, that is, no order is observed throughout the system. Figure 10 illustrates the dependence of the step width on the bulk temperature; the region next to the surface is ordered ($\theta_s = 1.69$), $a = 1.0$ and the interaction decays as $\epsilon_\ell \sim \ell^{-1}$.

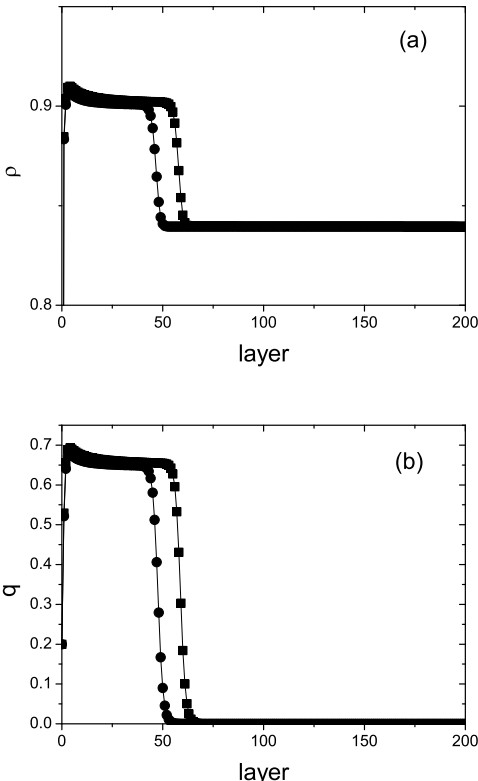

**Figure 9.** CS case. Profiles for: (**a**) occupancy and (**b**) orientational order parameter. Circles: $\epsilon_\ell \sim \ell^{-1}$; squares: $\epsilon_\ell \sim \ell^{-0.7}$. $\theta_s = 1.69$, $\theta_b = 1.80$ and $a = 1.0$. Lines are just guides to the eye.

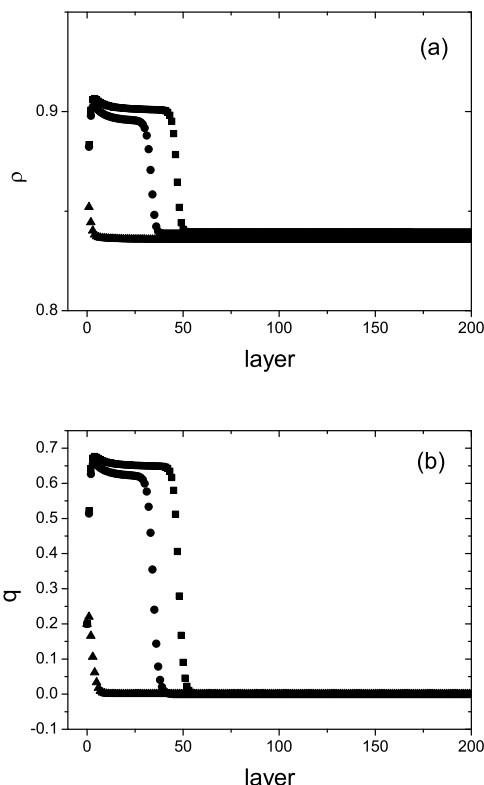

**Figure 10.** CS case. Profiles for: (**a**) occupancy and (**b**) orientational order parameter. $\epsilon_\ell \sim \ell^{-1}$; $\theta_s = 1.69$; $a = 1.0$; squares: $\theta_b = 1.80$, circles: $\theta_b = 1.82$, and triangles: $\theta_b = 1.90$.

We also verified the opposite case: a disordered bulk and an ordered surface, such that the bulk temperature is fixed at a high value ($\theta_b = 1.82$) and we keep decreasing the surface temperature, starting from $\theta_s = 1.65$. Appreciable changes in the step height and width are reached only at much lower values of $\theta_s$, of the order of 1.25. With these results (not shown here), we conclude that the steps shape is much more sensitive to the bulk than to the surface temperature.

Concerning the dependence of the step shape on $\mu$, we observed that the CS case responds similarly to what was reported for the PS case (Figure 6).

## 4. Discussion

### 4.1. Quenched Disorder

In order to compare the effects of disorder in related systems, we start this section by recalling a problem considered in our previous work [14]. There we had rods confined *inside* a disk-shaped cavity and all lattice sites were occupied, that is, we had a fixed concentration of particles. In order to allow for varying concentration, we introduce here permanent vacant sites: that is, a system with site *quenched* disorder, as opposed to *annealed* disorder studied so far. To our knowledge, there is no complete solution for the Potts model with site disorder in the literature [17], therefore we analyse this problem with an approximate argument.

We recall the expression for the free energy per column:

$$F = \sum_{\ell=0}^{\infty} \mu_\ell q_l - \sum_{\ell=0}^{\infty} \epsilon_\ell q_\ell q_{\ell+1} - \sum_{\ell=0}^{\infty} \epsilon_\ell (q_\ell)^2 + \frac{k_B T}{3} \sum_{\ell=0}^{\infty} [2(1-q_\ell)\ln(1-q_\ell) + (1+2q_\ell)\ln(1+2q_\ell) - 3\ln 3] , \quad (7)$$

where $\mu_\ell$ is an external potential which vanishes everywhere, except for $\ell = 1$ (next to the surface).

The corresponding recurrence relations for the order parameter are:

$$q_\ell = \frac{\mu_\ell}{2\epsilon_\ell} - \frac{1}{2}(q_{\ell+1} + q_{\ell-1}) + \frac{k_B T}{3\epsilon_\ell} \ln \frac{1+2q_\ell}{1-q_\ell}. \quad (8)$$

One way to introduce quenched dilution is to claim that the number or nearest neighbor sites is reduced by a factor $h$, where $0 < h < 1$. In that order, we replace all $\epsilon_\ell$'s by $h.\epsilon_\ell$'s in Equations (7) and (8), yielding the modified recurrence relations:

$$q_\ell = \frac{\mu_\ell}{2h\epsilon_\ell} - \frac{1}{2}(q_{\ell+1} + q_{\ell-1}) + \frac{k_B T}{3h\epsilon_\ell} \ln \frac{1+2q_\ell}{1-q_\ell}. \quad (9)$$

These last relations show that, in this case of quenched disorder, the introduction of vacancies by means of $h$ brings no qualitative novelty: first order transitions are expected as a function of the concentration (or $h$); moreover, the results obtained formerly [14] will be modified by rescaled $T$'s and $\mu_\ell$'s.

### 4.2. Annealed Disorder

The present study, which takes into account annealed disorder, produces a much richer dependence on the concentration, to be discussed next.

A first order transition is observed both in the PS and CS cases, as expected. The profiles depend on the chemical potential (or concentration) and on the strength of the interparticle interactions; nevertheless, the shape of the steps is determined primarily on the degree of ordering (temperature) in the bulk. All results are robust with respect to the strength of the homeotropic anchoring, they do not affect the profiles, except for the immediate neighborhood of the surfaces. Such behavior is not observed in simulations of similar systems [10–12], where profiles are strongly dependent on the anchoring condition. The reason for this discrepancy is that the mean-field approach intrinsically favors a preferential direction.

In the PS case, for vannishing $\mu$, we obtained profiles associated with different temperature values, all of them in the first order transition temperature range. The disordered solution in the low

temperature end of the coexiste window (unstable solution) has a step, reflecting the presence of the nearby surface; the disordered solution in the high temperature end of coexistence (stable solution) presents just a tiny step. Temperature is associated with the degree of order in the bulk; the step, which arises because of the anchoring at the nearby surface, can be destroyed for sufficiently high bulk disorder. We also characterize the isotropic nematic first order transition in terms as a complete orientational wetting scenario. The results for planar anchoring are presented; nevertheless, they are not very reliable due to the details of the mean-field approach.

In the CS case, the effect of decreasing interactions towards the bulk produces very interesting effects. For an ordered surface and a disordered bulk, the profile may present a step. A first observation regarding the steps is that their widths depend on the slope of the interaction profile, being wider for smoother profiles. We also showed that, similarly to what happens in the PS case, steps are destroyed when interactions in the bulk are weak enough. On the other hand, for a fixed degree of disorder in the bulk, enhancing the interactions' strength on the surfac, produces just a feeble effect on the magnitude of the order parameters and on the step width. These two observations lead us to conclude that the step height depends weakly on the surface temperature and their width is mainly determined by the bulk degree of disorder.

It is possible to compare our results for the bulk densities at the nematic isotropic with those of de Las Heras end Velasco [10]; they obtain a global packing fraction of the order or 0.29, to be compared to ours $\rho - \rho_0$ ($\rho_0 = 0.75$), which is approximately 0.15 for $a = 1.0$.

We also studied the effect of varying the chemical potential on the step shape. Increasing $\mu$ amounts to enhancing the overall concentration and produces wider steps. This behavior has also been observed in previous numerical studies on spherocylinders near flat [9] and curved walls [10].

Our motivation was to describe the distribution of colloids near surfaces, a three-dimensional problem. Solving the two-dimensional problem amounts to disregarding interactions between rods in neighboring planes. In a mean-field description, it consists of considering a smaller number of first neighbors. Having obtained the orientation profile on a given plane, we claim that this simplification reduces the three-dimensional $\rho_\ell$'s and $q_\ell$'s by a common factor. This affects the corresponding step widths and heights; nevertheless, the qualitative features of the profiles are the same. This is certainly true for flat surfaces and a good approximation for curved surfaces of large radii.

## 5. Conclusions

Motivated by recent experiments on colloidal particles with dimensions comparable to the confining volumes, we extend a model proposed in a recent work to describe the orientation of rods on planes bounded by straight or curved lines, considering the concentration as an extra parameter: it corresponds to the fraction of occupied sites. The particles are driven from a reservoir to the system by means of a chemical potential, so we allow for annealed disorder. They occupy sites on a semi-infinite lattice above a line or around a disk. We propose a lattice model to describe such systems and map it onto a three-state Potts model with vacancies, which is solved within a mean-field approach. In the curved case, the effect of the curvature is reproduced by layer dependent interparticle interactions. For both types of surfaces, we obtained the occupancy and the orientational profiles, which were studied under various conditions.

The main outcome of our study is to predict the non-smooth behavior of the profiles: the occurrence of steps both in the occupancy and in the orientational order parameters profiles. They may occur because the surfaces impose specific anchoring; nevertheless, their stability is primarily determined by the degree of order in the bulk. Steps like these have been also predicted in the literature, mainly in simulations, albeit with no systematic analysis. Furthermore, we believe that it may be of interest for applications in nanotechnology.

A shortcoming of our study is the weak influence of the anchoring on the overall profile. Moreover, we considered a fixed chemical potential throughout the sample; a more realistic procedure would be to model it as a decreasing function from the surface towards the bulk; this procedure would imply

a more complicated interpretation of the results. At this time, an extension of this problem is underway: biaxial particles are being considered.

**Funding:** This research received no external funding.

**Acknowledgments:** We thank M. J. de Oliveira for very useful discussions and comments.

**Conflicts of Interest:** The authors declare no conflicts of interest.

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
