# Peer review of "Ordering of Rods near Surfaces: Concentration Effects"

_crystals, doi:10.3390/cryst9050265_

Round 1

Reviewer 1 Report

Review of Ordering of rods near surfaces: concentration effects

crystals-461216

This is a very difficult to read manuscript.  At places it is so opaque that I was tempted to throw up my hands. Having gone through the manuscript, I believe that the results can be of interest to the larger community but the overall presentation of sections 1 and 2 needs to be made much clearer.

Specific comments:

The first two paragraphs make sense, but there is then a large discontinuity from paragraph 2 to paragraph 3.  Paragraph 3 starts defining the system.  It took me a while to conclude the colloidal LC are rectangular parallelepipeds.  What does “The particle dimensions are sizable compared to the surface curvature” mean? (A figure or something helps here). I see the spherical hole, but the lattice   as drawn on the next page varies from plane to plane.  I am lost on rows in the fourth paragraph.  It later seems that your three state Potts model is one state perpendicular to the surface and the two degenerate orthogonal directions perpendicular the other two states. A colloid goes onto some of the lattice sites and then obtains an orientation based on the model, equilibrium what happens, why and a longer explanation are all needed Rho and N don’t make sense to me and the picture does not enlighten me. Treating the planes as being independent might be OK at the equator, but if your model is what it appears to be at say an altitude of 450the colloid must penetrate a plane.  What is a noninteracting average and an interacting average? 

I do not understand the model, and there appears to be too little information for me to pull it all together

The author understands the model and has spent a great deal of time and effort on understanding it. This model was not explained to this referee well enough.  I’m afraid it will not be intelligible to many readers.

Author Response

Thank you very much for your clear coments, which indeed helped me improve the manuscript.

I wrote again the last two or three paragraphs of section 1, as well as sections 2.1 and 2.2. 

I think now it became easier to understand the model; following your suggestion I also

included a figure in order to illustrate how "... particles dimensions are sizeable compared..."

giving an example.

On paragraph of section 2.1, I meant the following: next to the "circle", the distance

between neighboring sites on the same row 2.pi.R/N must be larger than the mean width of the 

rectangle b.rho(0). On the upper rows there is more room for the the rods (rectangules), 

up to a point where they can start to rotate, and this happens at a distance r from the circle,

 where: 2.pi.r/N is larger than the mean length of the rectangule L.rho(r) (actually can be

L or L/2). 

Concerning the "non interacting average". I was mistaken: it is a term used

to label a step in the mean field calculations. It was corrected and replaced just by:

"average".

Reviewer 2 Report

Review of “crystals-461216”

Orientational ordering of colloidal rods is examined using the 3 state Potts model on a lattice in this work. Using mean-field approach the free energy is minimized with respect to order parameters. The resulting equations for the order parameters are solved numerically using iteration. It has been shown that a first order phase transition occurs between orientationally disordered and ordered phases upon increasing the chemical potential for both planar and spherical confinements. These results are justified with the curves of occupancy and orientational order parameters. Even if I find this work valuable, I do not support the publication of it. The reasons are followings:

1)      I am not convinced that the new results are relevant for people working in the field of colloids. The lattice calculations may have severe consequences for the density and orientational order parameter profiles, which is not present in continuum models. Therefore it would be useful to compare lattice calculations with some DFT and MC results, which were done for continuum models.

2)      This system is devised to model the colloidal LC, where the driving factor is the density for the phase transition. Here both the density and the temperature affect the phase behavior, which makes the problem more complicated. As a suggestion I would switch off the attraction to see clearly the effect of excluded volume interactions.

3)      It would be also useful to present the bulk phase diagram (the case of non-confinement), where the system undergoes first order orientational ordering transition. These results would allow us to make a connection between the bulk and confined systems.

4)      It is not mentioned in the text that what we see is a nematic wetting transition at the wall. This transition has been examined in several studies using MC and DFT methods.

5)      Why is the anchoring homeotropic and not planar? If we allow the anchoring to be planar, an even richer phase behavior can be observed such the biaxial nematic ordering in the vicinity of confinement.

6)      The length of the rod should have effect on the ordering properties. This is clear in the bulk where the more elongated rods exhibit nematic order at lower densities (or chemical potential).

7)      Some tiny comments: the system should have first order phase transition, i.e.“ both order parameters have first order phase transitions” in the text should be rewritten. In the introduction we can read that “In the past fifty years, a great deal of attention has been devoted to understand how liquid crystals are confined to small cells.”. It would sound better if “confined” is replaced with “ordered”. There are also some typo errors in the text like the “inside inside”.

In summary I would suggest a new study where the effect of hard body exclusion is considered only. In addition to this as the confinement is hard repulsive I would not restrict to surface ordering to be homeotropic. With such a study it would be possible to get closer to the experiments and simulations.

Author Response

I have a pdf file below.

Round 2

Reviewer 1 Report

This is a great improvement from the earlier version.

I have a few questions/comments on wording.

Line 21, first word.  Is surfaces a better choice than cells?

line 79, by row do you mean radial line.

I think equation 2 needs a reference- it is just dropped in.

Figure caption 5.  Here a=mu/kT, earlier (line 120) it is defined as a=mu/epsilon.  This must be corrected.  Figure 3, suggests a=mu/kT.

line 161 "the Potts... one direction." (no here).

Overall a much clearer presentation.

Author Response

Thank you very much for revising my manuscript again.

All modification requested in round 2 were made. 

Just one comment: above equation (2), there is a paragraph were I had  reference (16)  for eq. (2); may be it was "too far",  so, what I did was to displace this reference right close to eq. (2).

Reviewer 2 Report

The review of the revised manuscript “crystals-461216”

The new paper has been changed substantially even in this very short term. The author has added the bulk phase diagram, calculated the adsorption and considered the possible planar order. I must say the quality of the work has improved a lot, but I still think that the paper does not deserve to be published. The reason for this is that I am not convinced that the presented results are correct. My problems:

1)      I think that the Eq. (1) cannot be used for the present model, because it contains only one order parameter (q_l). This order parameter representation does not take into account that it is possible to get a biaxial nematic order in the vicinity of the wall. To make the theory correct, the new quantities in the trace of the matrix should be (-q_l/2+d_l, -q_l/2-d_l,q_l), which is also a traceless tensor. In this new formalism, the free energy must be minimized with respect to d_l, too. I am sure that the planar order at the wall will be more competitive with the homeotropic one.

2)      Author say: “I don't really understand what you mean by "switching off" the attraction, since all we have are excluded volume interactions.” If there is no attractive interaction between the particles, the configuration part of partition function does not depend on the temperature as the exp(-beta*U) is zero or one. This means that the temperature does not affect the phase transition, i.e. the system is athermal even in the presence of hard walls. Therefore the bulk phase diagram cannot be correct (Figure 3).  See for example the phase diagrams of the hard body fluids (hard sphere freezing, IN transition of hard rods, etc.) I think that the model potential has also some attractive interaction contribution. Therefore I strongly suggest to define the rod-rod and rod-wall interaction potentials in the paper.

3)      It is also not clear how the author determines the phase boundary between I and N phases. In continuum systems the equality of the pressures and the chemical potentials gives two equations for the coexisting densities. I think the author determined the regions of one solution (only isotropic, only nematic solutions) and the regions of two solutions (both isotropic and nematic solutions exist). If I am right, Fig. 3. is not a phase diagram.

4)      What is the expression for epsilon_l when eq. (4) is derived from eq. (3)? It is also not clear what is the epsilon_l in the planar calculation.

5)      How is the boundary condition specified in the calculations? I think we do not need any specific conditions at the walls (q(0) = 0.49, rho(0) = 0.9). Therefore Fig. 8 cannot be good by this setting.   

6)      “Moreover we considered a fixed chemical potential throughout the sample….”. I think the equilibrium situation gives the same chemical potential in the whole sample. Therefore the varying chemical potential would be a driving force of material transport which is not the aim of this work.

Author Response

none